# Advances in Pediatric HIV-1 Cure Therapies and Reservoir Assays

**DOI:** 10.3390/v14122608

**Published:** 2022-11-23

**Authors:** Priya Khetan, Yufeng Liu, Adit Dhummakupt, Deborah Persaud

**Affiliations:** 1Division of Infectious Diseases, Department of Pediatrics, School of Medicine, Johns Hopkins University, Baltimore, MD 21205, USA; 2W. Harry Feinstone Department of Molecular Microbiology and Immunology, Bloomberg School of Public Health, Johns Hopkins University, Baltimore, MD 21205, USA

**Keywords:** cure strategies, reservoir assays, intact provirus, pediatric HIV-1 infection

## Abstract

Significant advances in the field of HIV-1 therapeutics to achieve antiretroviral treatment (ART)-free remission and cure for persons living with HIV-1 are being made with the advent of broadly neutralizing antibodies and very early ART in perinatal infection. The need for HIV-1 remission and cure arises due to the inability of ART to eradicate the major reservoir for HIV-1 in resting memory CD4+ T cells (the latent reservoir), and the strict adherence to lifelong treatment. To measure the efficacy of these cure interventions on reservoir size and to dissect reservoir dynamics, assays that are sensitive and specific to intact proviruses are critical. In this review, we provided a broad overview of some of the key interventions underway to purge the reservoir in adults living with HIV-1 and ones under study in pediatric populations to reduce and control the latent reservoir, primarily focusing on very early treatment in combination with broadly neutralizing antibodies. We also summarized assays currently in use to measure HIV-1 reservoirs and their feasibility and considerations for studies in children.

## 1. Introduction

An estimated 38.4 million people were living with HIV-1 (PLWH) in 2021, of whom 1.7 million were children under the age of 15 [1]. Antiretroviral therapy (ART) is effective at suppressing HIV-1 replication such that the plasma viremia levels fall below the detection limits of clinical assays. The suppression can last for decades and enables PLWH to live without severe disease progression [2,3,4]. However, ART is limited in that it is unable to fully eliminate HIV-1 from the body due to the establishment of viral latency in resting memory CD4+ T cells [5,6,7,8,9]. The HIV-1 latent reservoir is the population of cells or anatomical sites that allow the persistence of replication-competent proviruses for life even in patients on effective ART [10,11,12,13]. These cells harbor HIV-1 proviruses capable of producing viral RNA and proteins following stimulation by antigens or activating agents, leading to production of infectious virions. Reservoirs are relatively stable as they are protected from ART and the immune system. Currently, research shows that the reservoir resides mostly in resting memory CD4+ T cells. However, there is increasing evidence that naïve CD4+ T cells can also harbor replication-competent provirus [5,8,12,13,14,15,16,17,18,19,20,21,22,23].

Studies in adults living with HIV-1 who were on durable effective ART for up to 7 years showed that the latent reservoir decays slowly with a half-life of 44 months, which indicated that a person would have to stay on ART for approximately 73.4 years to fully eradicate a reservoir size of approximately one million latently infected cells [5,7,16]. When ART is initiated early during acute HIV-1 infection, the size of the reservoir is smaller compared to when ART is initiated during chronic infection. However, even with early ART and a small reservoir size, viral rebound is observed upon treatment interruption, which attests to the major barrier HIV-1 reservoir cells pose to HIV-1 cure [24,25,26,27,28,29,30]. Typically, ART interruption results in a rebound in viremia within 2–4 weeks which supports the long-term persistence of an inducible reservoir [6,8,24,25,31,32].

In children living with perinatal HIV-1 infection, the size of the latent reservoir is not smaller than in adults if ART is started during chronic infection [33]. However, in perinatal infection, the latent reservoir becomes reduced over time with early effective ART initiated at <3 months of age and with very early ART started between birth and seven days of age in neonates with in utero infections [34,35,36]. Strict adherence to the prescribed regimen is required for sustained virologic suppression (SVS), and to prevent the selection of drug-resistant HIV-1. This is particularly difficult to achieve in children [37]. Several factors affecting adherence include, but are not limited to care giver adherence, ART side effects, and sustained accessibility to antiretroviral drugs (ARVs), along with the need for frequent follow-up appointments to assess ART efficacy [38,39]. Long-term use of ART is also associated with adverse side effects [40,41,42,43], and, importantly, stigma [44]. Altogether, these factors highlight the need for novel treatments in order to achieve ART-free remission and cure that can allow PLWH to not need lifelong ART for SVS, as highlighted below. Studies are underway towards finding new treatments to eliminate HIV-1 reservoirs for ART free remission and a cure of HIV-1, where ART can be stopped and SVS continued off ART [45,46].

ART-free remission in the case of perinatal HIV-1 infection can be defined as the ability to sustain virologic suppression in the absence of ART for one or more years, while maintaining normal CD4+ T cell levels and immune responses to childhood vaccines [47]. A few cases of ART-free remission have been reported in perinatal infections with very early ART (a girl—the Mississippi baby) [35], and with early ART (one girl-the French Adolescent) [34] and one boy (the South African boy) [36]. A subset of adults treated during acute infection in the VISCONTI Cohort experienced years of ART-free remission, also referred to as post-treatment controllers (PTCs) [48], which are distinct from elite controllers [49,50], thereby offering optimism towards long-term control of HIV-1 off ART. Notably, cases of HIV-1 cure have been reported to date in two adult men [51,52,53] and potential cure in one woman [54,55,56] through stem cell transplantation with CCR5 delta 32 homozygous cells, as part of treatment for malignancies they developed while on ART. More recently, cases of “natural” HIV-1 cures were identified in two women who were not on ART. In these two women, the persistent proviruses were found to be at exceedingly low levels and overwhelmingly defective [57,58]. These unique cases offer hope for HIV-1 cures and provide mechanistic insights into achieving this goal for more PLWH.

### 1.1. Pediatric HIV-1 Infection

HIV-1 can be transmitted from a mother to her infant via three routes: in utero, intrapartum, or postpartum through breastfeeding [59,60,61]. With the use of ART during pregnancy, mother-to-child transmission rates have fallen drastically compared to the pre-ART era. However, vertical HIV-1 transmission still occurs due to seroconversion of the mother during pregnancy, poor adherence to ART or no ART during pregnancy and the breastfeeding period [1,62].

### 1.2. Distinctive Features of Perinatal Infection

The immunologic environment of the fetus in utero is tolerogenic, anti-inflammatory [63,64], and is biased towards Th17, Treg and Th2 lineages instead of the pro-inflammatory Th1 lineage [64,65,66,67,68], with potential implications for reservoir size and its stability. The anti-inflammatory environment promotes low immune activation, and low expression of CCR5 on CD4+ T cells, thereby potentially limiting HIV-1 reservoir establishment [65,68]. Perinatal HIV-1 infections allow for rapid ART initiation (within 48 h of life to 3 months of age), which can pave the way for smaller reservoir size by restricting viral replication [68,69,70], while also providing an environment suitable for the introduction of immunotherapeutics that can potentially control the HIV-1 reservoir.

### 1.3. Maintenance and Expansion of the Reservoir

An important aspect of the HIV-1 reservoir is the contribution of clonal expansion of the cells that harbor the reservoir [71,72,73]. Clonal expansion is an umbrella term that refers to three mechanisms: homeostatic proliferation, antigen driven proliferation, and integration into or in close proximity to genes involved in cell growth [74,75,76,77]. Homeostatic proliferation occurs as a result of exposure to cytokines such as IL-7 and IL-15 [78,79,80,81,82]. Antigen driven proliferation occurs through repeated exposure to cognate antigens and has been shown to drive proliferation regardless of integration site [77]. HIV-1 proviruses have a propensity to integrate into active genes [83,84], such as genes associated with cell growth and proliferation (*STAT5B*, *BACH2*, *MLK1*), contributing to the maintenance of the reservoir over time [71,73,85] which altogether provide an unsurmountable barrier to HIV-1 eradication.

## 2. Therapies for HIV-1 Infection

Since the start of the HIV-1 epidemic, significant advances were made to find suitable life sustaining treatment options for PLWH, including children. With the discovery of the latent reservoir, the shift from lifelong treatment with ART to sustain control of HIV-1 to achieving ART free remission has been crucial. With this goal in mind, several novel interventions are under intense investigation, primarily in adults [45,86]. A few strategies are under investigation in the pediatric population for which very early ART with immunotherapeutics are the most feasible and promising. These interventions are described in Figure 1 and the major clinical trials in the pediatric population are summarized in Table 1 [86].

### 2.1. Current Interventions under Investigation for Pediatric HIV-1 Remission and Cure

#### 2.1.1. Very Early and Early Antiretroviral Therapy in Neonates to Reduce HIV-1 Reservoirs to Achieve Remission

The WHO recommends that ART in children should be initiated when the diagnosis of HIV-1 is confirmed, regardless of virologic and immunologic status as supported by studies showing the life-saving effects of ART with reductions in mortality and disease progression [69,87,88,89,90]. Very early antiretroviral therapy refers to ART initiation during the first few hours to days of life, but requires access to early infant testing with quick turn-around [35]. Studies on early and very early ART initiation in perinatal infection across different cohorts continue to show that ART initiation before six months of age is beneficial in reducing time to suppression of viremia and the reservoir size, which in a few pediatric cases has substantially delayed the time to rebound post treatment cessation [70,91,92,93,94,95,96,97,98,99,100,101,102,103]. This was first reported in the Mississippi baby, who experienced 27 months of ART-free remission starting at 18 months of age following triple antiretroviral drug initiation at 30 h of life [35]. In two other children, the French adolescent and the South African boy, with perinatally acquired HIV-1, post-treatment control occurred with early ART initiated at three and two months of life and stopped at five–six years and less than one year of life, respectively [34,36]. The details for the three cases of remission are described in Table 2. However, for most children treated from early infancy who experience markedly reduced reservoir size, virus rebound occurs within two–four weeks when ART is stopped [104,105].

#### 2.1.2. Broadly Neutralizing Antibodies for Use in Perinatal HIV-1 Infection to Achieve Remission

Recently, it was shown that in very early treated children, substitution of ART with a combination of broadly neutralizing antibodies (bNAbs) VRC01LS and 10-1074 was well tolerated [106], and the dual bNAb therapy permitted maintenance of virologic suppression for 24 weeks in 44% of such very early-treated children [107]. The results of this proof-of-concept study support the notion that reservoir reduction through early ART may enable long-term control with an immunotherapeutic intervention such as combination of bNAbs, although more studies are required.

bNAbs target conserved regions of HIV-1 Env epitopes, regardless of genetic variations within different HIV-1 subtypes leading to virus neutralization [108]. During the process of neutralization, the antibodies and HIV-1 virions form antigen-antibody complexes that promote immune clearance [109]. In one study of bNAb VRC01 in adults, 1 out of 14 adults achieved the goal of remaining off ART after 24 weeks of interruption (NCT02664415) [110]. With monotherapy, baseline resistance and selection of resistance are major limitations to the use of single bNAbs for ART-free remission and cure [111]. Hence, the use of dual or triple bNAbs, including bispecific antibodies, is being actively investigated in adult populations. Notably, a combination of two bNAbs, 3BNC117 and 10-1074, prolonged viral suppression for 20 or more weeks in 76% of the participants (NCT03526848) [112]. In addition, vector-based delivery of bNAbs via Adeno-Associated Virus (AAV) is being explored for long-term production of bNAbs [113]. In a proof-of-concept clinical trial (NCT03374202), all 8 adult participants were able to produce VRC07 with AAV delivery; 4 had stable concentrations of VRC07 production for up to three years, which suggests that gene therapy treatments may provide a long-term source of bNAbs [113,114]. Studies for use of bNAbs in HIV prevention in HIV-1 exposed neonates are ongoing and so far have shown that VRC01 was well tolerated and safe for use in this cohort (NCT02256631) [115]. The overall safety profile of bNAbs make them highly attractive therapies for HIV-1 prevention, remission, and cure for pediatric populations.

### 2.2. Interventions under Study to Achieve ART Free Remission and Cure

#### 2.2.1. Epigenetic and Provirus Targeted Therapies

The main purpose of epigenetic and provirus targeted therapies is to either target latent proviruses to express themselves through reversal of HIV-1 latency or to permanently silence the provirus, which are referred to as “shock and kill” or “block and lock” strategies, respectively [45]. Both strategies rely on understanding the establishment of HIV-1 latency. The “shock and kill” strategy has been studied in adults living with HIV-1 [116]. With this strategy, latency reversal agents (LRAs) can induce the latent provirus to become transcriptionally active and possibly lead to virus production, thereby allowing cells harboring the reservoir to be recognized and cleared by the immune system [116]. A comprehensive list of the LRA candidates in clinical trials in humans is maintained by the Treatment Action Group [86]. This list includes Vorinostat, Panobinostat and Romidepsin, which are histone deacetylase inhibitors (HDACi). No significant reductions in the reservoir size have been observed in clinical trials despite evidence of induction of HIV-1 transcription in vivo. This suggests that a combination of therapies will be required to eliminate the HIV-1 reservoir [45,116]. In a recent study, the eCLEAR study, it was shown that a combination of Romidepsin with a broadly neutralizing antibody given at the time of ART initiation led to a greater decrease in intact proviruses compared with ART only [117]. In general, the LRAs studied so far have been shown to be well tolerated [118,119,120]. However, none of the clinical trials based on the sixteen LRA candidates include children in the study population [86]. The HDACis have off-target effects, with activation of cellular RNA transcription of normal cells, and have not been studied in pediatric populations [121]. Of note, not all latent viruses can be induced and eliminated through this strategy; these uninduced reservoir cells may still contribute to viral rebound after ART interruption [122].

“Shock and Kill” strategies have shown to be challenging in eliminating HIV-1 reservoir cells. Hence, research on the block-and-lock strategy is emerging [123]. Proteins such as the Tat protein that are required for reactivation and do not have human homologs are attractive targets for intervention [124]. Tat recruits and activates RNA Polymerase II (RNAP II) for HIV-1 transcriptional elongation by recruiting a general RNAP II elongation factor, P-TEFb, which consists of CDK9 and Cyclin T1 [125,126,127]. Both CDK9 and Cyclin T1 are down-regulated during HIV-1 latency in primary resting memory CD4+ cells, and yet are required for the reactivation of latent HIV-1 proviruses [128,129]. As Tat is critical for the reversal of HIV-1 latency, studies are utilizing Tat inhibitors as an approach to permanently silence HIV-1 proviruses [124]. Didehydro-Cortistatin A (dCA), a natural steroidal alkaloid cortistatin A analog, is under investigation due to its Tat-suppressive function [130]. However, there are currently no therapeutic agents available to target the Tat protein.

#### 2.2.2. Immune System Targeted Treatment

Latency reversal may also be manipulated with drugs targeting immune pathways involved in the establishment and persistence of HIV-1 [116,123]. One such pathway is the JAK-STAT pathway, which is activated in HIV-1 infected macrophages and lymphocytes and is found to cause production of virions [131,132,133,134]. Two FDA-approved JAK-STAT inhibitors, Ruxolitinib and Tofacitinib, were found to be potent inhibitors of HIV-1 replication and virus reactivation in vitro [135]. A clinical trial of Ruxolitinib in ART treated adults (NCT02475655) showed that Ruxolitinib was well tolerated and associated with decreased markers of immune activation, but with no effect on total HIV-1 DNA or cell-associated HIV-1 RNA [136]. However, this medication is FDA-approved for use in individuals 9 years of age and older with intermediate or high-risk myelofibrosis and may be suitable in older children and youth living with perinatal HIV-1, if a decrease in the reservoir size is observed with these medicines [136].

Another attractive pathway to perturb HIV-1 latency involves the heat shock response (HSR) pathway which has multiple potential targets such as heat shock protein (HSP) 90, HSP20, HSP27, and heat shock factor 1 (HSF-1) that can be investigated [137,138,139]. HSPs are chaperone proteins that are involved in the production and folding of viral proteins to stabilize them [139]. A study testing Thiostrepton (TSR), a proteasome inhibitor in HIV-infected CD4+ T cell lines, revealed that the up-regulation of HSPs, which subsequently activated p-TEFb and the NF-KappaB (NF-κB) pathway, resulted in the reactivation of latent HIV-1 proviruses [140]. The effect of HSP90 on the reactivation of HIV-1 was further confirmed using the JLAT cell line [141]. Owing to these characteristics, HSPs may serve as targets for either “block-and-lock” or “shock-and-kill” strategies. 

Novel pathways that are involved in HIV-1 persistence are still being uncovered, including the mTOR (mammalian target of rapamycin) pathway which was first identified as relevant to HIV-1 persistence by a genome-wide shRNA screen [142]. In vitro evidence indicated that the inhibition of mTOR would result in suppression of latency reversal by blocking the transcription of Tat and Tat-mediated-elongation of HIV-1 [142]. The use of mTOR inhibitors to promote the silencing of HIV-1 was further explored in a clinical trial (NCT02429869) that utilized Everolimus in solid organ transplant recipients with controlled HIV-1 infection. Everolimus is an mTOR inhibitor that is frequently used as an immunosuppressant to prevent rejection of transplanted organs [143]. The study showed that the effect of Everolimus on cell-associated HIV-1 DNA and RNA concentrations was not evident across the cohort. However, in the participants who maintained trough levels consistently >5 ng/mL during the first two months of the treatment showed trends of decreasing cell-associated HIV-1 RNA concentrations even six months after cessation of Everolimus therapy. The authors showed that Everolimus reduced PD-1 expression, which might be an added benefit of the Everolimus therapy in HIV-1 infection. A few caveats to note: there was no control group in the study and the participants were on more than one immunosuppressive drugs that may have also impacted the observed results, implying the need for additional studies of the effects of Everolimus on basal levels of HIV-1 transcription and their implications for reservoir reduction [143].

HIV-1 evades the immune system by up-regulating the immune checkpoint ligand PD-L1 on the surfaces of antigen-presenting cells [144,145]. The high expression of PD-1 supports an integral role in HIV-1 latency [146]. PD-1 blockade has been shown to promote HIV-1 latency reversal in both in vitro and in vivo studies [146,147]. Hence, PD-1 and PD-L1 inhibitors are being explored as therapeutic targets [148].

Latency reversal agents that are attractive for studies in pediatric populations but not yet studied are Toll-like receptor (TLR) agonists. As described previously, the reactivation of HIV-1 from latency involves the activation of the NF-κB pathway [140]. TLR signaling pathways are known to trigger the NF-κB pathway [149], making TLR agonists potential LRAs with immune modulating function [150]. So far, agonists of TLR1/2 (in central memory cells), TLR5 (in central memory CD4+ T cells), TLR7 (in vitro using ART-suppressive donor cells), TLR9 (using CD4+ autologous cells) have all shown the ability to increase expression of latent HIV-1 in CD4+ T cell models [151,152,153,154]. In a recent clinical trial, Vesatolimod, a TLR-7 agonist, was reported to activate T cells and Natural Killer (NK) cells in adults on ART, but no significant change in plasma HIV-1 RNA concentrations was observed when compared to the placebo group (NCT02858401) [155]. In adult non-human primate models (NHP) the combination of TLR7 agonist Vesatolimod and an Env-targeting bNAb, PGT121, suppressed plasma viremia in 4 out of 8 animals after 24 weeks of ART discontinuation and one animal rebounded initially but later re-suppressed, while all 7 animals in the placebo group injected with saline rebounded after 17 weeks of ART interruption [156].

A recently described group of activators of the non-canonical NF-κB activation pathway, namely the second mitochondrial-derived activator of caspases (SMAC) mimetics, may also serve as potential LRAs [157]. SMAC mimetics activate the NF-κB pathway by inhibiting an upstream inhibitor, the cellular inhibitor of apoptosis protein 1 (cIAP-1). Activation of cIAP-1 represses the production of p52, a key component in the non-canonical NF-κB pathway [158]. A study showed that AZD5582 a SMAC mimetic, efficiently reversed latency in vivo in BLT humanized mice and also in adult rhesus macaques on continuous ART [157]. The same group later indicated that CD8+ T cell depletion enhanced the latency reversal effects of AZD5582, leading them to conclude that CD8+ T cells play a critical role in maintaining Simian Immunodeficiency Virus (SIV) latency [159]. Moreover, they showed that in infant rhesus macaques, the response pattern to AZD5582 differed, with some non-canonical NF-κB proteins such as NFKB2 and RELB showing less upregulation than in the adult NHP model [160]. These findings highlight the need for infant NHP studies in parallel with adult NHP studies to examine differences between pediatric and adult infections inducing HIV-1 reactivation [159].

Another area currently under investigation is the use of therapeutic vaccines to eliminate reservoir cells and maintain reservoir control off ART. A clinical trial (NCT02997969) in adults (18 to 40 years) showed that after participants received a clade C HIV DNA vaccine, an increase in V1/V2 antibody concentration and an increase in CD4+ T cell response to Env was observed [161]. In one study, transient decrease in HIV-1 reservoir size was seen with MVA and Fowlpox therapeutic immunizations in adolescents living with perinatal and non-perinatal infections [162,163,164]. A proof-of-concept clinical trial utilizing the HIVIS DNA vaccine followed by MVA-CMDR along with Human Papilloma Virus vaccine that contains a TLR-4 agonist in older children is completed and will examine effects of this intervention on reservoir size. The results from this trial will provide information on the use of therapeutic vaccines in perinatal HIV-1 infections to reduce reservoir size (NCT04301154).

#### 2.2.3. Gene Editing Based Treatment

Over the last decade, three cases of HIV-1 remission have been reported in patients that received Hematopoietic stem cell transplantation (HSCT) from donor with the rare CCR5 delta 32 mutation as a treatment for the malignancies developed while infected with HIV-1 [51,52,53,56]. These unique cases of long-term remission give researchers hope for eradication of HIV-1 using transplantation, but this approach remains challenging due to graft vs host disease, and the rarity of the CCR5 delta 32 allele in the human population [165]. Owing to these roadblocks, researchers are proposing to mimic the results seen in the HSCT cases with gene editing.

There are various strategies available to enable gene editing such as RNA interference (RNAi), Zinc finger nucleases (ZFNs), Transcription activator-like effector nucleases (TALENs), base editing, and clustered regulatory interspaced short palindromic repeats (CRISPR). ZFNs, CRISPR and TALENs rely on introducing double stranded breaks in the DNA which are repaired by either non-homologous DNA end joining (NHEJ) or homology-directed repair (HDR), which introduce indels and modify the function of the edited gene. RNAi involves the use of small interfering RNAs (siRNA) and short hairpin RNAs (shRNA) to bind to the viral transcripts and suppress translation [166]. Base editing uses proteins that can help target deaminases such as cytosine and adenine deaminases to the target site and modify the base without introducing double stranded breaks [167]. These strategies have been used to edit HIV-1 genes in primary CD4+ T cells and are being investigated for use in adults [86]. However, gene editing strategies need to be further evaluated for tolerability, feasibility, accessibility, ethical use, and efficacy, as well as potential undesirable outcomes such as off-target effects [166,167]. These factors need to be considered and more research will be required before these therapies can be approved for use in HIV-1 infections and subsequently adapted for use in pediatric HIV-1 infections.

In conclusion, the field of HIV-1 cure research is progressing with numerous potential interventions to eliminate and control the latent reservoir that precludes cure. The most promising strategies for pediatric populations include very early and early ART to reduce HIV-1 reservoir size paving the way for immunotherapeutic control of viremic rebound. The differences in the immune milieu and pathogenesis between pediatric and adult populations need to be considered as interventions are adapted for use in children.

## 3. Assays to Measure the HIV-1 Reservoir

Many assays exist to measure the HIV-1 reservoir in CD4+ T cells, but there are features that limit their broader use in clinical trials. It is therefore important to have an assay or a suite of assays that can reliably quantify the reservoir before and after treatment interventions to determine treatment efficacy. Assays that are ideal for the pediatric population would not require large specimen volume, and specifically measure the reservoir and not the predominant species of defective proviruses, characteristic of HIV-1 infection [10,13,14,168]. Table 3 and Figure 2 summarize the various assays used to measure the reservoir.

### 3.1. Classical Assays for HIV-1 Reservoir Measurement

#### 3.1.1. Culture Based Assays

##### Quantitative Viral Outgrowth Assay

The quantitative viral outgrowth assay (QVOA) was the first assay used to identify the latent reservoir for HIV-1 in resting memory CD4+ T cells in PLWH [5,6,8,33,169]. QVOA was used to demonstrate the incurable feature of HIV-1 and the need for lifelong ART [5,7] and is considered to be the gold standard for measuring the induced, infectious latent reservoir [12]. Briefly, purified resting CD4+ T cells are stimulated ex vivo in limiting dilution with mitogens in the presence of irradiated feeders and activated CD4+ T cell lymphoblasts for 14–21 days to quantify the frequency of intact induced infectious proviruses. Infectious virus is detected in the culture supernatant using an Enzyme Linked Immunosorbent Assay (ELISA) for the capsid protein p24, and the size of the latent reservoir is measured in infectious units per million (IUPM) using Poisson distribution [5,169]. In perinatal infection, ART initiated during chronic infection at a median of 8 years of age the size of the reservoir was found to be similar to adults at approximately 1.0 IUPM [33]. However, when ART is initiated in infancy and virologic suppression is sustained through early adolescence, the reservoir becomes smaller over time, and often at frequencies consistently undetectable at <0.1 IUPM [33,93,163,189]. The QVOA is challenging for use in clinical trials aimed at cure interventions, due to its complexity, cost, large blood volume requirement (180 mL in adults; 3–5 mL in infants and 15–50 mL in older children and adolescents) and labor-intensive nature. However, it still remains a prime minimal estimate assay for studying the replication-competent latent reservoir in PLWH, including children [12,22,33,93,163,168,170,190,191,192,193].

##### Tat/Rev Induced Limiting Dilution Assay

The Tat/Rev induced limiting dilution assay (TILDA) was developed to circumvent the complexity and large blood volume required for QVOA, and as a way to measure instead, the frequency of transcriptionally competent proviruses [171]. In the TILDA, 1–2 million total CD4+ T cells prepared in limiting dilution are stimulated with PMA and ionomycin for 12 h, followed by a nested reverse transcriptase quantitative polymerase chain reaction (RT-qPCR) for the multiply spliced HIV-1 RNA Tat/Rev transcripts. In comparing the frequency of latently infected CD4+ T cells between TILDA, and QVOA it was noted that the TILDA gave 48-fold higher values than QVOA in adult infections. Notably, the standard TILDA does not maximally detect the inducible reservoir in perinatal compared with adult infections, which can be overcome with the Enhanced TILDA that uses a combination stimulation approach with PMA, ionomycin and PHA and incubation for 18 h instead of 12 h [171,172]. The TILDA is less labor-intensive and less costly than QVOA; and is more discriminating by providing information on transcriptional competence of HIV-1 proviruses persisting under ART. The lower CD4+ cell requirement makes the TILDA feasible for application in clinical trials that will evaluate the effects of cure strategies on the transcriptionally competent latent reservoir with the caveat that that HIV-1 subtype specific primers are required [14].

### 3.2. Molecular Assays

#### 3.2.1. Quantitative PCR

The complexity and labor-intensive nature of the QVOA made it necessary to develop simpler assays to study the size of the HIV-1 reservoir. Among them was adaptation of quantitative PCR (qPCR) to measure HIV-1 DNA [168]. The qPCR is based on amplification of short amplicons in the conserved regions of HIV-1 genes such as *pol* or *gag* [173]. The results are interpreted by generating a standard curve from plasmid controls and calculating the relative quantities of HIV-1 DNA copies [168,170,174]. Due to the small amplicon size, PCR assays are sensitive, but single amplicon-based PCRs cannot differentiate between intact and defective proviruses, and therefore are an imprecise measure of reservoir size, since the latter dominate the proviral landscape [194]. Intact HIV-1 proviral genomes lack fatal small and large deletions, insertions, premature stop codons or hypermutations [13,179] while defective HIV-1 proviral genomes are not capable of producing infectious virions due to large deletions spanning one or more regions of the genome, insertions, frameshift mutations, hypermutations mediated by APOBEC3G/F, and mutations that affect viral fitness [194,195].

Additionally, these assays also quantify non-integrated HIV-1 present as 2LTR circles, which in perinatal infection, are present in high concentrations in the first two years of treatment in infancy [189], further leading to overestimates of reservoir size. Importantly, however, single amplicon-based HIV-1 DNA PCR assays provide a near maximal estimate of the frequency of HIV-1infected cells [10,168] and have been used extensively to study the decay of cell-associated DNA in both adult and perinatal infections and over long-term ART [34,35,36,69,95,98,99,100,103,192,196,197,198,199,200]. qPCR methods are hindered by the need for plasmid controls from which the HIV-1 DNA load is derived, which introduces variability across laboratories [168]. In addition, the number of cells analyzed is calculated from amount of DNA added to the reaction.

#### 3.2.2. Alu PCR

To counter the inability of standard PCRs to differentiate between integrated and non-integrated forms, a different type of qPCR was developed which makes use of the *Alu* regions found in the human genome and targets *GAG-LTR* from HIV-1 to help specifically detect integrated HIV-1 DNA [176,177]. Similar to a qPCR, a standard curve is required for quantification of results which can be inefficient as not all Alu sequences will be in close proximity to the HIV-1 genome causing errors in detection. Therefore, there is need for a correction factor to account for this issue [173,175]. Alu-PCR is less studied in perinatal infections [95,103].

#### 3.2.3. Single-Plex Droplet Digital PCR 

Droplet digital PCR (ddPCR) has revolutionized HIV-1 DNA quantitation as it permits an absolute quantification of HIV-1 provirus without the need for standard curves [178]. ddPCR uses droplets in oil emulsion to form thousands of nano-sized droplets such that each droplet is its own PCR reaction, giving a more precise quantitative signal. The ddPCR targets conserved regions in the *5′ LTR* and either *gag* or *pol* in the HIV-1 genome. Upon comparison with QVOA, it was found that in adults the frequency of infected cells detected by ddPCR was 300-fold higher than the size of latent reservoir measured by QVOA [170], highlighting the preponderance of defective HIV-1 proviruses persisting on ART. One of the advantages of ddPCR over qPCR is that the number of cells analyzed can be easily determined via a PCR reaction that targets the housekeeping gene *RNase P30 (RPP30)* which is run alongside the HIV-1 specific PCR [178]. Initiation of ART early in perinatal infections has proven to be beneficial in lowering the proviral loads. However, sometimes the proviral loads are very low and need a more sensitive method than qPCR to detect 1–2 copies/million cells [45,47,201]. The ddPCR allows this for this quantification due to its high resolution and sensitivity across HIV-1 subtypes, which are critically important for cure studies underway in perinatal infections [70,91,92,93,96,101,189,190,202].

### 3.3. Recently Developed Assays for Reservoir Measurement

#### 3.3.1. Molecular Assays

##### Intact Proviral DNA Assay 

The intact proviral DNA assay (IPDA) is a multi-plex ddPCR assay designed to quantify the latent reservoir. In the IPDA, two regions of the HIV-1 genome are targeted: the packaging sequence (*psi*, *Ψ*) upstream of the *gag* and the Rev Response Element (*RRE*) in the *env* gene. These two regions were chosen based on near full-length genome sequence (nFGS) analyses and bioinformatics where it was shown that any deletions in these regions indicate a high probability that the HIV-1 genome is defective. The IPDA also includes a double quencher probe for hypermutations near the *env* region (most variable region) [179]. A strong correlation between the size of the reservoir measured by QVOA and intact proviral copies per million cells as inferred by the IPDA was found, suggesting that this molecular assay provides a good estimate of the latent reservoir size [203]. Measurement of proviral load in perinatal infections using single-plex ddPCR or qPCR so far have been beneficial in providing the near total concentration of HIV-1 which is important [70,91,196]. However, to develop and assess cure interventions for the pediatric population dissection of the reservoir dynamics is imperative. The IPDA is able to achieve this due to its ability to differentiate between intact and defective proviruses and its requirement for low blood volume. Hence it serves as an important tool for studying the reservoir in pediatric infections where collection of large blood volumes may not always be feasible. However, the use of short amplicons in IPDA to determine the integrity of the genome can lead to over-estimation of the number of intact proviruses if the defects in the genome do not overlap the regions covered by the primers. The IPDA described here is only validated for HIV-1 subtype B, thereby limiting its use for study in non-subtype B HIV-1 infection.

##### Cross Subtype Intact Proviral DNA Assay

Recently, a modified version of the conventional 2-target IPDA was developed and subsequently adapted for multiple HIV-1 subtypes. The modified version of the IPDA targets five regions (5T) in the HIV-1 genome namely, 5′ end of *pol*, *tat*, 3′ end of *pol*, *gag* and *env* regions. A special feature of this assay includes also determining the number of T cells analyzed via a third PCR reaction (as a part of the *RPP30* PCRs) that detects the region of the T-cell receptor (TCR), which is deleted in mature thymocytes (*delta D*) [184]. The 5T-IPDA was optimized for subtype B. However, the highest disease burden is in Africa where subtype B is not the predominate subtype. Subtype differences may lead to differences in sequences and regions that are predictive of intactness of the genome. Therefore, developing an IPDA that can be used for several subtypes can circumvent this problem. The 5T-IPDA served as a springboard for the development of the cross-subtype IPDA which targets only the 3′ *pol*, *gag* and *env* regions. The cross-subtype PCR is reported to differentiate intact and defective genomes in participants with living with HIV-1 subtypes A, B, C, D and CRF_01 [185], a critical advance for the field, if validated bioinformatically as was done by the group that developed the conventional 2-target IPDA [179]. The cross subtype IPDA was recently examined in Kenyan infants with HIV-1 Subtype A infection [185].

### 3.4. Flow Cytometry

#### HIV-1 Flow

HIV-1 flow is a recently developed technique that involves stimulation of cells with PMA and Ionomycin followed by flow cytometry to determine the translational competency of HIV-1- infected cells and their immunophenotype [204]. In HIV-1 Flow, isolated and stimulated CD4+ T cells are labeled with two anti-p24 antibodies to detect cells infected with translationally competent HIV-1. The cells are also labelled with a series of cell surface marker antibodies to determine CD4+ T cell subsets, activation, and exhaustion [204]. The combination of antibodies in HIV-1 flow allows for simultaneous probing of cells that are positive for p24 and their phenotype making it a potentially useful tool to study the reservoir and especially in pediatric populations where blood volume is limited.

### 3.5. Proviral Landscape Analysis

#### 3.5.1. Near Full-Length Individual Proviral Sequencing 

Sequencing approaches are extremely helpful in characterizing the reservoir of HIV-1 since they can help differentiate between defective and intact proviruses, provide information about clonal expansion, integration sites, HIV-1 diversity, and immune escape, allowing for assessment of the efficacy of immunotherapeutic interventions. Near full-length individual proviral sequencing (FLIP-seq) involves the amplification of the provirus using an outer PCR with HIV-1 specific primers to generate near full-length HIV-1 followed by nested PCRs, either as a near full-length inner 9 kb product or in segments as subgenomic regions and next generation sequencing (NGS) is performed, and the sequences generated are assembled to yield the proviral landscape [11,194,195,205]. The FLIP-seq can determine whether a provirus is genetically intact or defective and the locations of the defects contained by the provirus, along with hypermutation, and assessment of the contribution of clonal expansion to reservoir maintenance [11,76,186]. Near full-length sequencing has been used in perinatal infections showing paucity of intact proviruses over long-term early ART in perinatal infections [93,97] and more recently preferential deletion of intact proviruses with very early ART of neonates [202]. However, as the FLIP-seq assay depends on an initial outer 9 kb long distance PCR, it can underestimate the amount of intact provirus due to the inefficiency of the outer 9 kb PCR which needs to be considered when using sequencing approaches for quantitative analyses of reservoir size [183].

#### 3.5.2. Quadruplex Quantitative PCR

The quadruplex quantitative PCR assay also known as Q4PCR, was developed using in-silico analysis of hundreds of proviral sequences from the Los Alamos Database. It consists of performing a limiting dilution of HIV-1 DNA on which a *gag* specific qPCR is performed to determine limiting dilution, followed by a near full-length outer PCR. The products of the outer PCR are then subjected to a multi-plex qPCR reaction where four regions of the HIV-1 genome: *env*, *pol*, *psi* (*Ψ*) and *gag* are interrogated [182]. The nested inner PCR followed by next generation sequencing are only done on samples that show a positive qPCR reaction for two or more regions of the HIV-1 genome. This approach of sequence verification increases the probability of the assay to detect truly intact proviruses by eliminating the proviruses that are classified as intact in the qPCR but may have defects in the regions not overlapping the primers [181]. The Q4PCR allows for a less sequence-intensive approach, and therefore less costly approach to distinguish intact and defective proviruses.

#### 3.5.3. Matched Integration Site Analysis and Proviral Sequencing 

Matched Integration site analysis and proviral sequencing (MIP-seq) is newer technique that involves amplifying the proviruses using multiple displacement amplification (MDA) at a single genome level followed by NGS to allow for integration site analysis at the single provirus level and their linkages. MIP-seq therefore provides information about the intactness of the provirus as well as its chromosomal integration site [85,187]. Sequencing techniques are able to characterize the full proviral landscape, including integration sites, sequence intact proviruses and are complement to the cruder and more feasible assays such as IPDA to characterize the reservoir [179].

#### 3.5.4. Parallel HIV-1 RNA, Integration Site and Proviral Sequencing 

The parallel HIV-1 RNA, integration site, and proviral sequencing (PRIP-seq) assay is a modification of the FLIP-SEQ and MIP-SEQ assays. The PRIP-seq has multiple components, in addition to providing sequence intactness of a provirus (FLIP-seq) and its integration site data (MIP-seq), it is also able to study the transcriptional competence of the provirus in parallel. Therefore, at the proviral level, one can investigate how the integration site, proviral sequence and translational competence intermix and play a role in the stability of proviral reservoir cells [188].

## 4. Conclusions

In summary, the field of HIV-1 cure therapeutics is rapidly evolving for both pediatric and adult populations living with HIV-1, and for which the testing landscape is advancing. In current clinical trials involving the pediatric population, plasma HIV-1 RNA is the biomarker for viral suppression and HIV-1 DNA is the biomarker used to determine size of the reservoir, using RT-qPCR, qPCR and ddPCR assays. The qPCR and ddPCR quantify the near maximal number of infected cells giving a total HIV-1 DNA readout and have been thoroughly validated across multiple HIV-1 subtypes in past clinical trials and have shown to be highly sensitive in cases where the HIV-1 proviral load was exceedingly small. Further validation of assays such as the IPDA and TILDA will improve our understanding of the intact and inducible reservoir in perinatal HIV-1 infections. Altogether, each reservoir measurement approach will enhance our understanding of HIV-1 persistence and its relevance to HIV-1 cure research and its therapeutics, particularly for immunotherapeutics. Special considerations are needed for clinical trial endpoints for the pediatric population where blood volume is limited and where therapies can lead to very low infected cell frequencies that are still largely dominated by defective proviruses, and yet not signifying HIV-1 remission or cure.

## Figures and Tables

**Figure 1 viruses-14-02608-f001:**
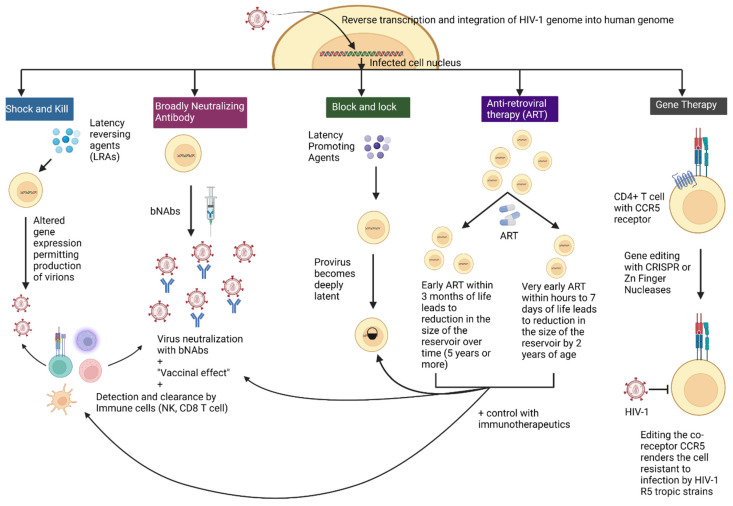
Various cure strategies to eradicate HIV-1 infection. Upon infecting the cell, HIV moves to the nucleus where it inserts its cDNA genome into the host genome to form the proviral reservoir. The reservoir is persistent and quiescent and poses a barrier to cure. It needs to be eradicated or reduced substantially to achieve cure or ART free remission. There are several cure interventions currently under study namely “shock and kill”, “block and lock” and gene editing. “Shock and kill” involves the use of latency reversing agents that forces the provirus out of latency and allows it to become transcriptionally active, and with some agents produce virions that can then allow for clearance by the immune system. Broadly neutralizing antibodies (bNAbs), in combination with “shock and kill” strategies may facilitate such immune -mediated clearance. The “block and lock” approach is a more recent approach and involves using latency promoting agents that modify the epigenetic environment of the provirus to keep it in a state of deep latency such that it is not reactivated. Gene editing utilizes different strategies to modify the CCR5 receptor on CD4+ T cells making the cells resistant to infection by HIV-1 R5 tropic strains. Very early and early ART in perinatal infection reduce HIV-1 reservoirs over time and in combination with immune strategies may promote ART-free remission and cure. (Figure created with BioRender.com).

**Figure 2 viruses-14-02608-f002:**
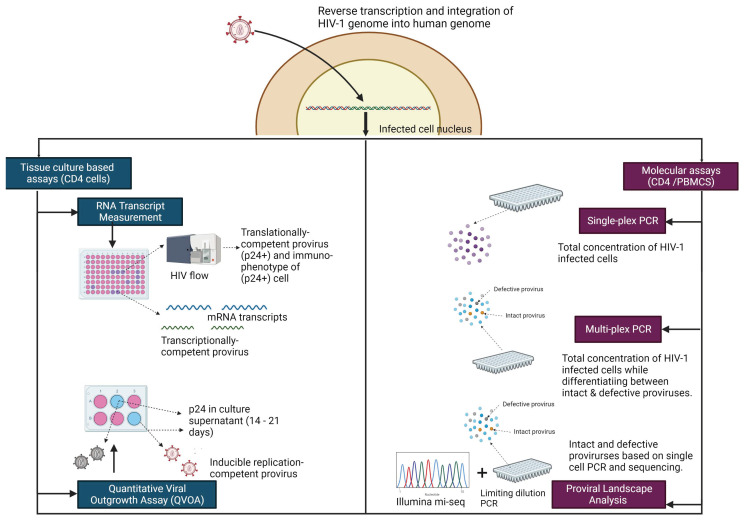
Assays used to measure the HIV-1 reservoir. HIV enters the cell nucleus and forms the proviral reservoir by inserting the cDNA of its genome into the host genome. The proviral reservoir is complex and made of several different species which can be measured using different assays that fall under two categories: Tissue culture-based assays and Molecular assays. Tissue culture-based assays (on the left) can measure the transcriptional competence of the provirus (TILDA) or the replication competence and infectivity of the provirus (QVOA). Molecular assays (on the right) can provide the total concentration of infected cells (single-plex ddPCR) or differentiate between intact and defective proviruses (multi-plex ddPCR). Molecular assays can also be used to study the dynamics of the proviral landscape using near full-length PCR and next generation sequencing technology. All of these assays provide key information about the reservoir dynamics and can be useful in developing and measuring the efficacy of cure interventions. (Figure created with BioRender.com).

**Table 1 viruses-14-02608-t001:** Clinical trials towards HIV-1 remission in children. (adapted from the Treatment Action Group (TAG HIV Science) website) [86].

Trial/Protocol Name	Trial Number	Age Range for Eligibility	Intervention	Country	Goal
IMPAACT 2008	NCT03208231	0 to 12 weeks of age	Combination of Early ART and VRC01	Botswana, Brazil, Malawi, Zimbabwe	Early clearance of HIV-1 infected cells in infancy
IMPAACT 2015	NCT03416790	13 to 24 years of age	Long-term ART	United States	Central nervous system reservoir characterization
IMPAACT 2039	In development	3–12 years of age	HIVconsvX vaccines with/without triple bNAbs	To be determined	Safety, Immunogenicity efficacy of HIVconsX vaccines with/without bNABs to control viremia off ART
IMPAACT 2028	NCT05154513	1 year and older	Follow up study of HIV persistence biomarkers in remission and cure trials (received Early or Very Early ART)	Botswana, Brazil, Haiti, Kenya, Malawi, South Africa, Tanzania, Thailand, Uganda, United States, Zimbabwe	Reservoir and immune biomarker profile following cure interventions
P1107	NCT02140944	12 months and older	Cord blood transplantation with CCR5 delta 32 stem cells	United States	HIV cure
P1115	NCT02140255	Up to 10 days of life	Very early ART of neonates with/without BNABs	Argentina, Brazil, Haiti, Kenya, Malawi, Puerto Rico, South Africa, United States, Tanzania, Thailand, Uganda, Zambia, Zimbabwe	ART free remission
EIT (Early Infant HIV Treatment)	NCT02369406	0 to 56 days of life	Very Early ART	Botswana	ART free remission
LEOPARD (Latency and Early Neonatal Provision of Antiretroviral Drugs Clinical Trial)	NCT02431975	Up to 48 h of life	Very Early ART	South Africa	ART free remission
Tatelo Study	NCT03707977	96 weeks to 7 years	Early ART + combination bNAbs	Botswana	Safety and efficacy of dual bNAb VRC01LS and 10-1074 to control viraemia off ART
HIV-Netherlands Australia, Thailand Research Collaboration	NCT00476606	1 day to 20 years	Early ART	Thailand	Evaluate immunological and clinical outcomes of early ART
HVRRICANE Trial	NCT04301154	9 years or older	ART + HIVIS-DNA vaccine + MVA-CMDR boost with or without TLR-4 agonist	South Africa	Safety and effects of using primer boost vaccine regimens with/without TLR 4 agonist
Antiretroviral Regime for Viral Eradication in Newborns	NCT02712801	0–1 day of life	Very Early ART	China	HIV Cure

**Table 2 viruses-14-02608-t002:** Comparison of the three cases of ART-free remission in perinatal HIV-1.

Profiles	Mississippi Baby (2013) [35]	French Adolescent (2017) [34]	South African Child (2019) [36]
**Intervention**	Very Early ART	Early ART	Early ART
**Age at ART initiation**	30 h	3 months	2 months
**Sex**	Female	Female	Male
**Duration of ** **intervention**	18 months	5.8–6.8 years	40 weeks
**Age at remission ** **detection**	23 months	18.6 years	9.5 years
**Duration of remission**	27.6 months	>12 years	8.5 years
**Biomarker profile: ** **HIV DNA ** **(Log_10_ copies per million PBMCs)**	Nondetectable (<0.43)	Detectable (2.2)	Detectable (0.69)
**HIV-1 Serostatus**	Seronegative	Seropositive	Indeterminate
**Low level Viremia**	Undetectable	Detectable	Detectable
**Inducible reservoir**	Not detectable	Detectable	Detectable
**HIV subtype**	B	H	C

**Table 3 viruses-14-02608-t003:** Different assays for reservoir measurement.

Assay	Measure	Advantages	Disadvantages
Culture based assay			
Quantitative Viral Outgrowth Assay (qVOA) [5,12,22,168,169,170]	Replication competent infectious virus	Minimal estimate of the latent reservoir, reproducible	Long turnaround time (21 days), laborious, not all intact proviruses are induced, large cell number required, expensive
Tat-Rev Inducing Limiting Dilution Assay (TILDA) [14,171,172]	Transcriptionally competent virus	Shorter turnaround time than QVOA, no RNA extraction required, sensitive, reproducible, specific to HIV	Cannot differentiate between transcripts from intact and defective proviruses, not all intact proviruses are induced
Molecular assay			
Quantitative PCR (qPCR) [10,168,170,173,174]	Total HIV-1 proviral DNA	Low volume required, cost effective, short turnaround time, high throughput	Overestimates size of reservoir, cannot differentiate between intact vs defective and integrated vs non- integrated forms, relative quantification via standard curves
Alu PCR [173,175,176,177]	Total integrated HIV-1 proviral DNA	Can differentiate between integrated and non-integrated forms, cost effective, short turnaround time, high throughput	Overestimates size of reservoir, cannot differentiate between intact vs defective, relative quantification via standard curves
Droplet digital PCR (ddPCR) [14,170,178]	Total HIV-1 proviral DNA	Low volume required, high sensitivity, high throughput, short turnaround time, cost effective, more accurate than qPCR due to absolute quantification	Overestimates size of reservoir, cannot differentiate between intact vs defective and integrated vs non- integrated forms
Intact proviral DNA assay (IPDA) [14,179,180]	Intact and defective proviruses	Low volume required, high sensitivity, high throughput, short turnaround time, cost effective, can differentiate between intact, 5′ defective, 3′ defective and hypermutated	Overestimates size of the reservoir, cannot differentiate between integrated vs non-integrated forms, subtype B specific
Quadraplex quantitative PCR (Q4PCR) [181,182,183]	Intact and defective proviruses	Provides information on genetic intactness of provirus, low volume required, sensitive, can differentiate between intact and defective proviruses, high throughput compared to conventional near full- length genome sequencing	Cannot differentiate between integrated vs non-integrated forms, subtype B specific, relies on initial long-distance PCR so not quantitative as a standalone assay, low throughput, expensive
5 Target-Intact Proviral DNA Assay (5T-IPDA) [184]	Intact and defective proviruses	Low volume required, high sensitivity, high throughput, short turnaround time, cost effective, can differentiate between intact and defective proviruses	Overestimates size of reservoir, cannot differentiate between integrated vs non-integrated forms, complex analysis, require 2 sets of controls, subtype B specific
Cross Subtype-IPDA (CS-IPDA) [185]	Intact and defective proviruses	Similar to 5T-IPDA and can work across different subtypes A, B, C, D, CRF_01	Similar to 5T-IPDA
Proviral Landscape Analysis			
Near full-length individual proviral sequencing (FLIP-seq) [11,76,183,186]	Intact and defective proviruses	Provides information on genetic intactness of provirus	Low throughput, expensive, complex, time consuming, cannot amplify intact proviruses at the same frequency as defective proviruses due to inefficiency of the initial long-distance PCR
Matched integration site and proviral sequencing (MIP-seq)) [85,183,187]	Intact and defective proviruses; integration site of proviruses	Provides information on integration site and clonal expansion, genetic intactness of provirus	Similar to FLIP-seq
Parallel HIV-1 RNA, integration site and proviral sequencing (PRIP-seq) [188]	Intact and defective proviruses, integration site and transcriptional competence of provirus	Provides information on the genetic intactness, integration site, clonality and transcriptional competence of a provirus	Similar to FLIP-seq

## Data Availability

Not applicable.

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
