# Peer review of "Advances in Pediatric HIV-1 Cure Therapies and Reservoir Assays"

_viruses, 2022, doi:10.3390/v14122608_

Round 1

Reviewer 1 Report

This a review manuscript from Khetan et al, where advances in HIV cure interventions are presented in the pediatric population. In addition, the review describes multiple HIV reservoir assays that could be used in clinical trials to evaluate the efficacy of these cure interventions. The review is well written, very clear and easy to follow. Already published work on HIV reservoir in cohorts’ studies and current clinical intervention in the pediatric population are described. The literature and bibliography presented is very extensive.

Some comments could improve the review:

-       When authors describe the viral rebound after ART interruption even in early treated individuals, it would be interesting to include Colby et al Nature Medicine 2018 describing viral rebound in Fiebig I individuals.  (line 42)

-       Since early ART is a good intervention to limit the size of the reservoir, I would suggest to add some sentences regarding the work already published in the impact of early ART in the size of the HIV reservoir in the pediatric population. There are limited reviews on this topic (HIV reservoir + pediatric) and it could be interesting for the reader to know the impact of early ART without any additional cure intervention. Some important papers would be: Bitnun et ak CID 2014, Ananworonich et al AIDS 2014, Martinez-Bonet et al CID 2015, Van Zyl et al J int Dis 2015, McMannus et 2016, Tagarro et al JAIDs 2018, Massanella et CID 2020, among many others, including references from the author’s group.

-       It would be also important to acknowledge the work from other pediatric cohorts: EPIICAL and Thai pediatric cohorts, which also have published important work on the HIV cure pediatric agenda.

-       For the reservoir quantification, I would suggest to include mainly articles on pediatric population that have used these techniques. For example, the authors mention Cassidy et al, iScience 2022 for cross-clade IPDA assay, a study that has been done in pediatric samples. Citing specific pediatric work, the reader could benefit from “tricks” used by other investigators to overcome the limitations of working with pediatric samples (i.e limited cell available or other specific parameters that need to be taken into account to work in children samples.)

o   Luzuriaga JID 2014, Persaud AIDS 2012, Rainwater-Lovett PlosOne 2017: QVOA

o   Martinez-Bonet CID 2015, Garcia-Broncano et al., Sci. Transl. Med 2019: ddPCR

o   Ananworonich, AIDS 2014: integrated HIV DNA

o   Massanella et al CID 2020: TILDA, integrated HIV DNA

o   Garcia-Broncano et al., Sci. Transl. Med 2019: near full length sequencing

o   Etc, etc

Minor

-       Line 105: add T-cells to “normal CD4 T-cells levels….”

Author Response

Thank you for your comments and suggestions. Please see the attachment.

Reviewer 2 Report

In the manuscript “Advances in Pediatric HIV-1 Cure Therapies and Reservoir Assays” by Khetan et al. the authors summarize HIV cure therapies and assays to estimate the size of the HIV reservoir with an added focus on how these treatments and measurements apply to HIV infected children.  The manuscript is for the most part well written and researched and would be of interest to readers of Viruses.  I have the following comments for the authors to improve the manuscript prior to publication.

Major comments:

1. In the introduction the authors state that “the latent reservoir is the stably integrated, infectious proviral DNA in resting memory CD4+ T cells that is able to persist under ART”. But according to their definition they then say that it could include other cells.  Overall, I find it repetitive to have an introduction and a list of definitions.  These should all be explained fully in the introduction text instead of in two places. My recommendation is to remove the list of definitions and partial subheadings (Pediatric HIV-1 infection, etc…) and expand the introduction to properly introduce these terms.

2. For section 2 and Figure 1. In my opinion a very critical approach to cure is missing, that being the modification of cells by gene therapy (lentiviral transduction of antiviral genes and/or CCR5 gene editing). Given the successes of the four to five individuals cured of HIV with CCR5 d32/d32 cell transplant, this strategy is perhaps even more promising than some of those shown in Figure 1 and discussed in this section (ie block and lock) for reaching an HIV cure in both adults and children. The authors site a clinical trial (P1107) in Table 2 for this approach but it is not presented or discussed in this review.  There has been a lot of progress with gene editing, RNA interference therapies and other technologies that can be used to generate HIV resistant cells and potentially lead to a functional cure for HIV that should be mentioned.

3. Lines 477 to 488:  It is not clear how this technique is used exactly to measure the size of the reservoir. Could the authors provide some more details about this method?  For example, what kind of cells are specifically looked at for reactivation of HIV based on the list of cell surface markers the authors provide?  How does it distinguish between productively and latently infected cells? 

4. The manuscript would greatly benefit from a section discussing the use of the different techniques they describe in clinical trials.  Which of these techniques has been used to evaluate reservoir size in some key clinical trials and what are their advantages and weaknesses?  Perhaps a Table listing some key clinical trials and the techniques used to quantify the reservoir would help inform the reader about the current status of these assays or at least an extra column in Table 1 indicating which of these assays was used in those trials where applicable.

5. The conclusion is rather brief and not particularly informative. Could the authors expand on their conclusions and provide some perspectives about where they see the field going over the next decade?

Minor comments:

Line 54: “follow-up appointments to access ARVs” do the authors mean to say “to assess the effects of ARVs?”  they have already mentioned accessibility in the point before.

Lines 73-75: This sentence is unclear, please reword, particularly the end: “More recently, cases of “natural” HIV-1 cures were identified in two women in whom the persistent proviruses are at exceedingly low levels and are defective, occurring the absence of long-term ART”

Line 121-123: The use of the word unique here is not correct as there are many other instances when ART can and has been initiated early.

Lines 229-231:  It is unclear to me what the authors mean by “conservatively encoded”, do they mean that Tat is highly conserved?  Results from PMID: 25808207 suggest that Tat is not among the more conserved HIV proteins.

Lines 412-420 and Lines 421-437:  It should be pointed out that although Alu PCR does only detects integrated proviruses and ddPCR has some advantages over qPCR, they still cannot discriminate between defective and intact proviruses.

Line 426 “the 5’ the LTR” should be “the 5’ LTR”

Line 440 to 459 and 460 to 476:  Again, while these assays offer an improvement by looking at regions that may commonly lead to defective provirus it should be pointed out that this method is still limited in its ability to discriminate between defective and intact proviruses as several other regions of the provirus could have mutations/deletions/insertions that result in defective proviruses.

Author Response

(The authors gave the same response as above.)

Round 2

Reviewer 1 Report

The authors addressed correctly all the comments.

Author Response

Thank you for reviewing the paper and providing your suggestions. They were very helpful in improving the manuscript. 

Abstract - remove "and" in line 19: "...focusing on very early treatment and in combination with broadly neutralizing antibodies."

Response: Thank you for the suggestion. We have made the requested change and marked it with a comment.

Table 1 in the row starting with IMPAACT 2028 - correct "prolife" to "profile" in last column: "Reservoir and immune biomarker prolife following cure interventions"

Response: Thank you for catching the error. We have rectified it to spell profile and marked it with a comment.

Line 282 - remove "a" in this sentence: "...Tat protein that are required for reactivation and a do not have human homologs...

Response: Thank you for identifying the typo. We have removed it and marked it with a comment.

Lines 389-391 - edited for clarity: "These unique cases of long-term remission give researchers hope for eradication of HIV-1 using transplantation, but is difficult due to graft vs host disease, and the CCR5Δ32 is rare in the human population." Change to "These unique cases of long-term remission give researchers hope for eradication of HIV-1 using transplantation, but this approach remains challenging due to graft vs host disease, and the rarity of the CCR5Δ32 allele in the human population."

Response: Thank you for the suggestion. We have made the requested change and marked it with a comment.

Line 437 - move references [5,7] to before the word "and" - "...and the need for lifelong ART and [5,7] is considered..."

Response: Thank you for the suggestion. We have moved the references forward and marked the word ‘and’ with a comment.

Reviewer 2 Report

The authors have addressed all of my concerns.

Author Response

Thank you very much for your suggestions and comments. They were helpful in improving the manuscript. 

Abstract - remove "and" in line 19: "...focusing on very early treatment and in combination with broadly neutralizing antibodies."

Response: Thank you for the suggestion. We have made the requested change and marked it with a comment.

Table 1 in the row starting with IMPAACT 2028 - correct "prolife" to "profile" in last column: "Reservoir and immune biomarker prolife following cure interventions"

Response: Thank you for catching the error. We have rectified it to spell profile and marked it with a comment.

Line 282 - remove "a" in this sentence: "...Tat protein that are required for reactivation and a do not have human homologs...

Response: Thank you for identifying the typo. We have removed it and marked it with a comment.

Lines 389-391 - edited for clarity: "These unique cases of long-term remission give researchers hope for eradication of HIV-1 using transplantation, but is difficult due to graft vs host disease, and the CCR5Δ32 is rare in the human population." Change to "These unique cases of long-term remission give researchers hope for eradication of HIV-1 using transplantation, but this approach remains challenging due to graft vs host disease, and the rarity of the CCR5Δ32 allele in the human population."

Response: Thank you for the suggestion. We have made the requested change and marked it with a comment.

Line 437 - move references [5,7] to before the word "and" - "...and the need for lifelong ART and [5,7] is considered..."

Response: Thank you for the suggestion. We have moved the references forward and marked the word ‘and’ with a comment.